# Metazoan zooplankton in the Bay of Biscay: 16 years of individual sizes and abundances combining ZooScan and ZooCAM imaging systems.

## Authors

Grandremy Nina[1]*, Bourriau Paul[1], Daché Edwin[2], Danielou Marie-Madeleine[3], Doray Mathieu[1], Dupuy Christine[4], Forest Bertrand[5], Jalabert Laetitia[6], Huret Martin[7], Le Mestre Sophie[7], Nowaczyk Antoine[8], Petitgas Pierre[9], Pineau Philippe[4], Rouxel Justin[10], Tardivel Morgan[10], Romagnan Jean-Baptiste[1]*.

## Correspondence

grandremy.n@gmail.com, jean.baptiste.romagnan@ifremer.fr

## Affiliations

[1] DECOD (Ecosystem Dynamics and Sustainability), IFREMER, INRAE, Institut Agro, Nantes, Centre Atlantique - Rue de l'Ile d'Yeu - BP 21105 - 44311 Nantes Cedex 03, France.

[2] Unité Biologie et Ecologie des Ecosystèmes marins Profonds, Laboratoire Environnement Profond, Ifremer Centre Bretagne - ZI de la Pointe du Diable - CS 10070 - 29280 Plouzané, France.

[3] Unité DYNECO-PELAGOS, Laboratoire d'Ecologie Pélagique, Ifremer Centre Bretagne - ZI de la Pointe du Diable - CS 10070 - 29280 Plouzané, France.

[4] BIOFEEL, UMRi LIENSs, La Rochelle University / CNRS, 2, rue Olympe de Gouges, 17000 La Rochelle, France.

[5] Laboratoire Hydrodynamique Marine, Unité RDT, Ifremer Centre Bretagne - ZI de la Pointe du Diable - CS 10070 - 29280 Plouzané, France.

[6] Sorbonne Université, Institut de la Mer de Villefranche, 06230 Villefranche-sur-mer, France.

[7] DECOD (Ecosystem Dynamics and Sustainability), IFREMER, INRAE, Institut Agro, Centre Bretagne - ZI de la Pointe du Diable - CS 10070 - 29280 Plouzané, France.

[8] UMR CNRS 5805 EPOC – OASU, Station Marine d'Arcachon, Université de Bordeaux, 2 Rue du Professeur Jolyet, 33120 Arcachon, France.

[9] Departement Ressources Biologiques et Environnement, Ifremer Centre Atlantique - Rue de l'Ile d'Yeu - BP 21105 - 44311 Nantes Cedex 03, France.

[10] Laboratoire Détection, Capteurs et Mesures, Unité RDT, Ifremer Centre Bretagne - ZI de la Pointe du Diable - CS 10070 - 29280 Plouzané, France.

* These authors contributed equally to this work.

## Abstract

This paper presents two metazoan zooplankton datasets obtained by imaging samples collected on the Bay of Biscay continental shelf in spring during the PELGAS integrated surveys, over the 2004-2019 period. The samples were collected at night, with a WP2 200 µm mesh size fitted with a Hydrobios (back-run stop) mechanical flowmeter, hauled vertically from the sea floor to the surface with a maximum depth set at 100 m when the bathymetry is deeper. The first dataset originates from samples collected from 2004 to 2016, imaged on land with the ZooScan and is composed of 1,153,507 imaged and measured objects. The second dataset originates from samples collected from 2016 to 2019, imaged on board the R/V *Thalassa* with the ZooCAM and is composed of 702,111 imaged and measured objects. The imaged objects are composed of zooplankton individuals, zooplankton pieces, non-living particles and imaging artefacts, ranging from 300 µm to 3.39 mm Equivalent Spherical Diameter, individually imaged, measured and identified. Each imaged object is geolocated, associated to a station, a survey, a year and other metadata. Each object is described by a set of morphological and grey level based features (8 bits encoding, 0 = black, 255 = white), including size, automatically extracted on each individual image. Each object was taxonomically identified using the web based application Ecotaxa with built-in, random forest and CNN based, semi-automatic sorting tools followed by expert validation or correction. The objects were sorted in 172 taxonomic and morphological groups. Each dataset features a table combining metadata and data, at the individual object granularity, from which one can easily derive quantitative population and communities descriptors such as abundances, mean sizes, biovolumes, biomasses, and size structure. Each object's individual image is provided along with the data. These two datasets can be used combined together for ecological studies as the two instruments are interoperable, or as training sets for ZooScan and ZooCAM users. The data presented here are available in the SEANOE dataportal: https://doi.org/10.17882/94052 (ZooScan dataset, Grandremy et al., 2023c) and https://doi.org/10.17882/94040 (ZooCAM dataset, Grandremy et al., 2023d).

## Keywords

Zooplankton, ZooCAM, ZooScan, Bay of Biscay, imaging, PELGAS surveys.

# 1 Introduction

Metazoan planktonic organisms, hereafter referred to as zooplankton, encompass an immense diversity of life forms, which have successfully colonized the entire ocean, from eutrophic estuarine shallow areas to oligotrophic open ocean, from sunlit ocean to hadal depth. Their body sizes span five to six orders of magnitude in length, from µm to tens of meters (Sieburth & Smetacek, 1978). Zooplankton plays a pivotal role in marine ecosystem (Banse, 1995). It transfers the organic matter produced in the epipelagic domain by photosynthesis to the deeper layers of the ocean (Siegel et al., 2016), by producing fast sinking aggregates (Turner, 2015), and by diel vertical migration (Steinberg et al., 2000; Ohman & Romagnan, 2016). Zooplankton therefore participates in mitigating the anthropogenic carbon dioxide build up in the atmosphere responsible for climate change. Moreover, zooplankton is an exclusive trophic resource for commercially important fish during their larval stage, where a shift in zooplankton species or phenology can have dramatic effects on recruitment (i.e. North Sea cod, Beaugrand et al., 2003). In addition, it is a major trophic resource for adult planktivorous small pelagic fish, known as forage fishes (Van der Lingen, 2006). Recent studies suggest that zooplankton dynamics may have a significant effect on small pelagic fish population dynamics and individual body condition (Brosset et al., 2016; Menu et al., 2023), and therefore impact wasp-waist ecosystem based fisheries and fisheries dependent socio-ecosystems, worldwide (Cury et al., 2000).

Despite zooplankton being of such global importance in both climate change effects on ecosystems and management of fisheries (Chiba et al., 2018; Lombard et al., 2019), it is still technically difficult to monitor, with respect to other marine ecological compartments. Zooplankton biomass, diversity and spatio-temporal distributions cannot be estimated from spaceborne sensors as phytoplankton's does (Uitz et al., 2010), and zooplankton commercial exploitation data do not exist yet, as fish data does. One noticeable exception is the CPR surveys network that enables zooplankton data generation at spatio-temporal scales resolved enough to study climate change and diversity related zooplanktonic processes (Batten et al., 2019). Yet, generating zooplankton data often requires dedicated surveys at sea, specific sampling instruments and trained taxonomic analysts. Moreover, besides actual observation, modelling zooplankton remains a challenging task due to the diversity of traits such as life forms, life cycles, body sizes and physiological processes exhibited by zooplankton (Mitra & Davis 2010; Mitra et al., 2014). However, over the past two decades the development of imaging and associated machine learning semi-automatic identification tools (Irisson et al., 2022) have greatly improved the capability of scientists to analyse long (Feuilloley et al., 2022), high frequency (Romagnan et al., 2016), or spatially resolved (Grandremy et al., 2023a) zooplankton time series, as well as trait based data (Orenstein et al., 2022). Imaging and machine learning have particularly enabled the increased development of combined size and taxonomy zooplankton ecological studies (i.e. Vandromme et al., 2014; Romagnan et al., 2016; Benedetti et al., 2019). Yet, use of these machine learning tools is not trivial because these require abundant, scientifically qualified, sensor specific, training image data (i.e. learning set and test set, Irisson et al., 2022), and complex hardware and software setups (Panaïotis et al., 2022). One good example of such image dataset is the ZooScanNet dataset (Elineau et al., 2018), which features an extensive ZooScan (Gorsky et al., 2010) imaging dataset usable as a training set for ecologists as well as for imaging and machine learning scientists.

The objective of this paper is to present two freely available zooplankton imaging datasets, originating from two different instruments, the ZooScan (Gorsky et al., 2010), and the ZooCAM (Colas et al., 2018). These

datasets originate from the PELGAS integrated survey in the Bay of Biscay (Doray et al., 2018a), a continental

shelf ecosystem supporting major European fisheries (ICES, 2021). Combined together, these datasets make up a

16-years time series of sized and taxonomically resolved zooplankton, along with context metadata allowing the

calculation of quantitative data, covering the whole Bay of Biscay continental shelf, from the French coast to the

continental slope, and from the Basque country to southern Brittany, in spring. These datasets can be used for

ecological studies (Grandremy et al., 2023a), machine learning studies, and modelling studies.

## 2 Methods

### 2.1 Sampling

Zooplankton samples were collected during the successive PELGAS (PELagique GAScogne) integrated

surveys carried out over the Bay of Biscay (BoB) French continental shelf, every year in spring from 2004 to 2019

on board the R/V *Thalassa*. The aim of this survey is to assess small pelagic fish biomass and monitor the pelagic

ecosystem to inform ecosystem based fisheries management. Fish data, hydrology, phyto- and zoo-plankton

samples and megafauna sightings (marine mammals and seabirds) are concomitantly collected to build long-term

spatially resolved time series of the BoB pelagic ecosystem. The PELGAS sampling protocols combine day-time

en-route data collection (small pelagic fish and megafauna), with night-time, depth integrated hydrology and

plankton sampling at fixed points. Detailed PELGAS survey protocols can be found in Doray et al. (2018a) and

Doray et al. (2021). The PELGAS survey datasets providing hydrological, primary producers, fish and megafauna

data are available as gridded data in the SEANOE dataportal (Doray et al., 2018b) under the following link:

https://www.seanoe.org/data/00422/53389/.

The number of zooplankton samples across years varied between 41 (2005) and 64 (2019), due to

adjustments in the sampling strategy and weather conditions, for 889 zooplankton samples collected in total. From

2004 to 2006, samples were collected in the southern Bay of Biscay until the Loire estuary only (Fig. 1). Sampling

was carried out in vertical tows during night time using a 200-µm mesh size WP2 net, generally from 100 m depth

(or 5 m above the seabed) to the surface. In 2004 and 2005, the targeted maximum sampling depth was 200 m. In

2004, fifteen samples were collected deeper than 100 m, among which eleven were deeper than 120 m; in 2005,

twenty samples were collected deeper than 100 m, among which thirteen were deeper than 120 m. Before 2014,

the sampled water volume was estimated by multiplying the cable length by the net opening surface (0.25 m²)

whereas since 2014, the net was equipped with a Hydrobios back-run stop flowmeter. The samples originating

from 2004 to 2016 surveys were preserved in 4% formaldehyde (final concentration) and analysed on land in the

laboratory with the ZooScan, while since 2016 they were analysed live on board with the ZooCAM.

### 2.2 Sample processing and analyses

### 2.2.1 Digitization with the ZooScan

Preserved samples were digitized with the ZooScan (Gorsky et al., 2010), a flatbed scanner generating

16-bit gray-level high-resolution images (2400 dpi, pixel size: 10.56 µm, image size: 15×24 cm equivalent to

14 200×22 700 pixels). It is well suited for the imaging of preserved organisms ranging in size from 300 µm to

several centimeters. The ZooScan is run by the custom made, ImageJ based, ZooProcess software which generates

one single large image for each scan that contains up to 2000 organisms depending on the size of the imaged

organisms.

Prior to digitization, the seawater and formaldehyde solution was filtered through a 180 µm mesh sieve
into a trash tank, under a fume hood. The organisms were then gently but thoroughly rinsed with freshwater over
the tank, in the sieve. They were then size-fractionated with a 1 mm sieve, into organisms larger and smaller than
1 mm size fractions. This size splitting step is recommended when using the ZooScan to address the possible
under-representation of large objects bias caused by the necessary subsampling. Each size fraction was subsampled
separately with a Motoda splitter to obtain two subsamples containing 500-1000 objects for the large organisms
size fraction, and 1000-2000 objects for the small organisms size fraction. Each subsample was imaged after
manual separation of objects on the scanning tray, to mitigate the number of overlapping objects as recommended
in Vandromme et al. (2012). Overall, 699 samples were digitized following this protocol, corresponding to 1397
scans (one sample was not size fractioned as it did not contained organisms larger than 1 mm).

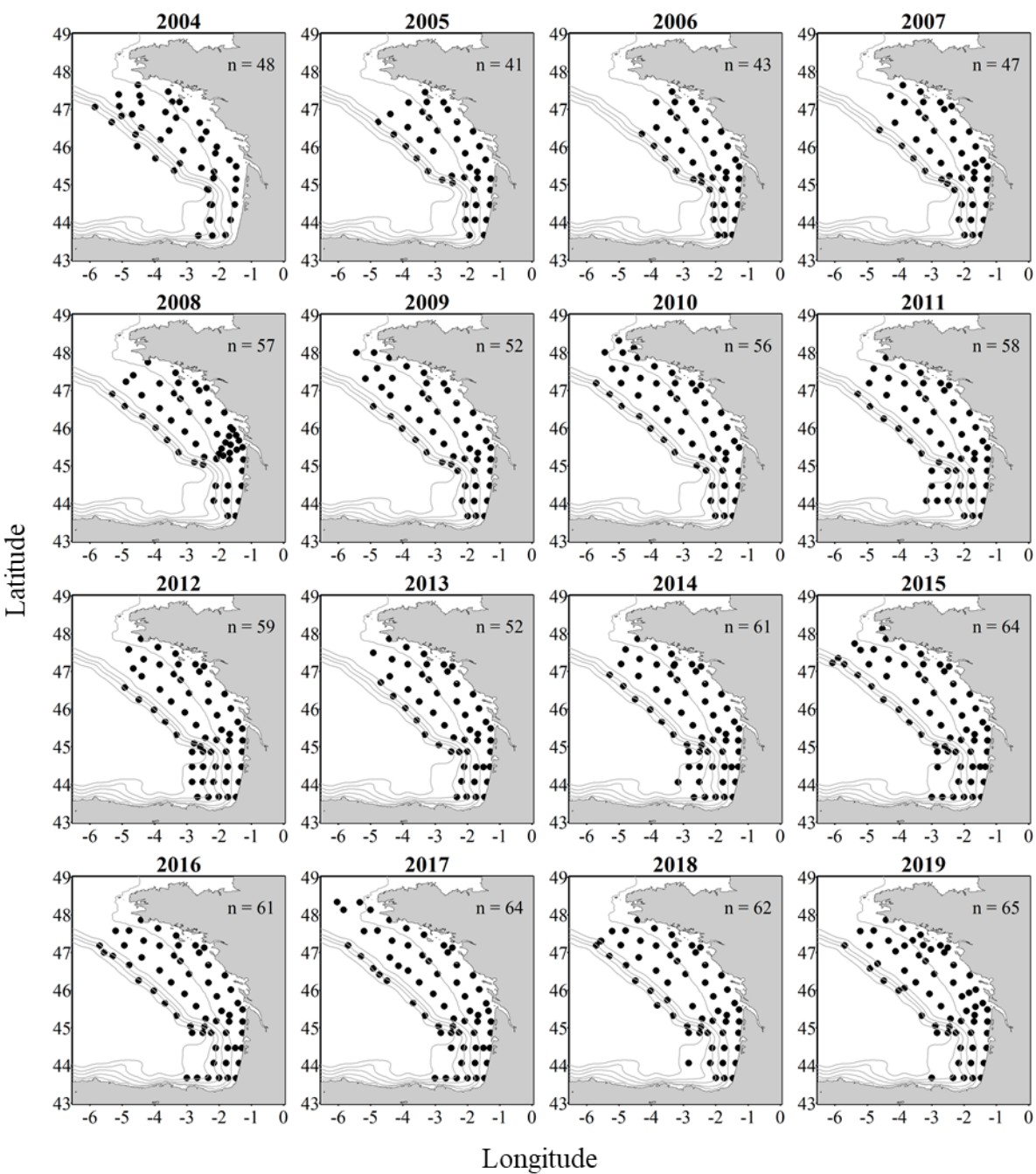

Figure 1: Metazoan zooplankton sampling locations during the PELGAS cruises in the Bay of Biscay from 2004 to 2019. The years with the poorest coverage are 2005 and 2006 with 41 and 43 sampling stations respectively; and the years with the best coverage are 2015, 2017 and 2019 with 64, 64 and 65 sampling stations respectively.

### 2.2.2 Digitization with the ZooCAM

The ZooCAM is an in-flow imaging instrument, designed to digitize preserved as well as live zooplankton samples, on board, immediately after net collection (Colas et al., 2018). The ZooCAM features a cylindrical transparent tank in which the zooplankton sample is mixed with filtered seawater. Depending on the richness of the sample, and the subsampling (if necessary), the volume of seawater can be adjusted between 2-7 litres. The organisms were pumped at a 1L.min$^{-1}$ from the tank to a flowcell inserted between a CCD camera (pixel size: 10.3 µm) and a red LED flashing device where they were imaged at 16 fps. Given the flowcell volume, the size of the field of view, the imaging frequency and the flowrate, all the seawater volume containing the organisms was imaged (Colas et al., 2018). Before all the initial volume was imaged, the tank and the tubing were carefully and thoroughly rinsed with filtered seawater to ensure the imaging of all the organisms poured in the tank. For each sample, the ZooCAM generates a stack of small size (~1 Mo) raw images that are subsequently analysed with the ZooCAM software. Depending on the initial water content of the tank and the rinsing, a ZooCAM run can generate up to 10k raw images from which the individual organism vignettes will be extracted. A ZooCAM run on a live sample often generates up to 5000-10000 vignettes of individual organisms. It is very important to subsample the initial samples with a dichotomic splitter (here a Motoda splitter), to get subsamples with a quantity of objects that reduce the risk of imaging overlapping objects, and avoid any dependency to the water volume imaged to reconstruct quantitative estimates of zooplankton as the initial and rinsing volume are variable. Overall, 190 samples were digitized live on-board with the ZooCAM.

### 2.3 Images processing

Both instruments generate grey level working images (8 bit encoding, 0 = black, 255 = white). In both cases, image processing consisted in (i) a "physical" background homogenization by subtracting an empty background image to each sample image (1 for ZooScan, and as many as raw images for ZooCAM), (ii) a thresholding of each raw image (threshold value: 243 for ZooScan, 240 for ZooCAM), (iii) the segmentation of each object imaged. The ZooProcess software was set to detect and segment objects with an area equal or larger than 631 pixels, whereas the ZooCAM software was set to detect objects with an area equal or larger than 667 pixels, which in both cases equals 300 µm ESD, or a biovolume of 0.014 mm$^3$ (using a spherical biovolume model, Vandromme et al., 2012).

Morphological features were then extracted on each detected object. Features generated by the ZooScan are defined in Gorsky et al. (2010) and those generated by the ZooCAM are defined in Colas et al. (2018). ZooScan images were processed with ZooProcess v7.39 (04/10/2020) open source software. ZooCAM images were processed with the proprietary ZooCAM custom made software which uses the MIL (Matrox Imaging Library, Dorval, Québec, Canada) as the individual object processing kernel. Each detected object was finally cropped from the working sample images, and saved as a unique, labelled vignette, in a sample specific folder along with a sample specific single text file containing the objects features arranged as a table with objects arranged in lines and features in columns.

### 2.4 Touching objects

The ZooProcess features a tool that enable the digital separation of possible touching objects in the final image dataset, for each sample. As touching objects may impair the estimations of abundances and size structure

(Vandromme et al., 2012), remaining touching objects were searched for on the individual vignettes from the ZooScan and digitally manually separated with the ZooProcess separation tool to improve the quality of further identifications, counts and size structure of zooplankton. The ZooCAM software does not offer such a tool.

## 2.5 Taxonomic identification of individual images

All individual vignettes from both instruments were sorted and identified with the help of the online application Ecotaxa (Picheral et al., 2017), as two instrument-specific separated sets. Ecotaxa features a Random Forest algorithm (Breiman, 2001) and a series of instruments specific tuned spatially sparse Convolutional Neural Networks (Graham, 2014) that were used in a combined approach to predict identifications of unidentified objects. First, an automatic classification of non-identified individual vignettes into coarse zooplankton and non-zooplankton categories was carried out. In both cases (ZooScan and ZooCAM), Ecotaxa hosted instrument specific image datasets, previously curated and freely available, that were used as initial learning sets. These initial classifications were then visually inspected, manually validated or corrected when necessary, and taxonomically refined when possible. After a few thousand images were validated in each project, they were used as dataset specific learning sets to improve the initial coarse automatic identifications. This process was iterated until all the individual vignettes were classified into their maximum reachable taxonomical detail. A subsequent quality check of automatic taxonomic identifications has been realized in a two-step process: a first complete review (validation and / or correction) of all individual automatic identifications was done by GN and RJB; then, trained experts (JL and NA) reviewed and curated the ZooScan and the ZooCAM datasets, respectively, at the individual level. Although some identification errors may still remain in the datasets, we consider this double check process as sufficient to provide taxonomically qualified data.

## 2.6 Intercalibration of the two instruments

The two datasets are usable separately. However, considered together they build a 16 years long spatio-temporal time series. A comparison study was done to ensure these datasets are homogeneous and can thus be combined for ecological studies (Grandremy et al., 2023b). All the zooplankton samples from year 2016 (61 sampling stations over the whole BoB continental shelf) were imaged with both instruments. In brief, all non-zooplankton and touching objects images were removed from the initial datasets. Then, the interoperable size range was determined with an assessment based on the comparison of Normalized Biovolume – Size Spectra (NB-SS) for each instrument. This size interval ranges between [0.3-3.39] mm ESD. Finally, the zooplankton communities as seen by the ZooScan and the ZooCAM were compared by taxa and by station using 27 taxonomic groups. Poorly represented taxa as well as non-taxonomically identified objects were not taken into account in the zooplankton variables computation and in community structure analyses. Both instruments showed similar NB-SS slopes for 58 out of 61 stations; depicted equivalent abundances, biovolumes and mean organisms' sizes, as well as similar community composition for a majority of sampling stations. They also estimated similar spatial patterns of the zooplankton community at the scale of the Bay of Biscay. However, some taxonomic groups showed discrepancies between instruments, which originates from the differences in sample preparation protocols before the image acquisition, the imaging techniques and quality, and whether the samples were imaged live or fixed. For example, the mineralized protists (here, Rhizaria) dissolve in formalin and are considered underestimated in preserved seawater samples (Biard et al., 2016). Also, the random orientation of objects in the ZooCAM flow cell leads to a loss of taxonomic identification accuracy due to the difficulty to spot the specific features needed for the

identification (Colas et al., 2018; Grandremy et al., 2023b). This is particularly acute for copepods, where the
ZooScan seems to provide better identification capabilities to experts, as the organisms are imaged in a lateral
view most of the time whereas the ZooCAM often images them in a non-lateral, randomly-oriented view,
preventing the visualisation of specific features. A detailed discussion about how to explain the discrepancies
between the ZooScan and the ZooCAM can be found in Grandremy et al. (2023b). We assume that the two
presented datasets build a single, 16 years long spatio-temporal time series of abundances (Fig. 2) and sizes of
zooplanktonic organisms (Fig. 3), from which biovolumes, biomasses, Shannon index (Fig. 4), and zooplankton
community size structure can be derived (Vandromme et al., 2012).

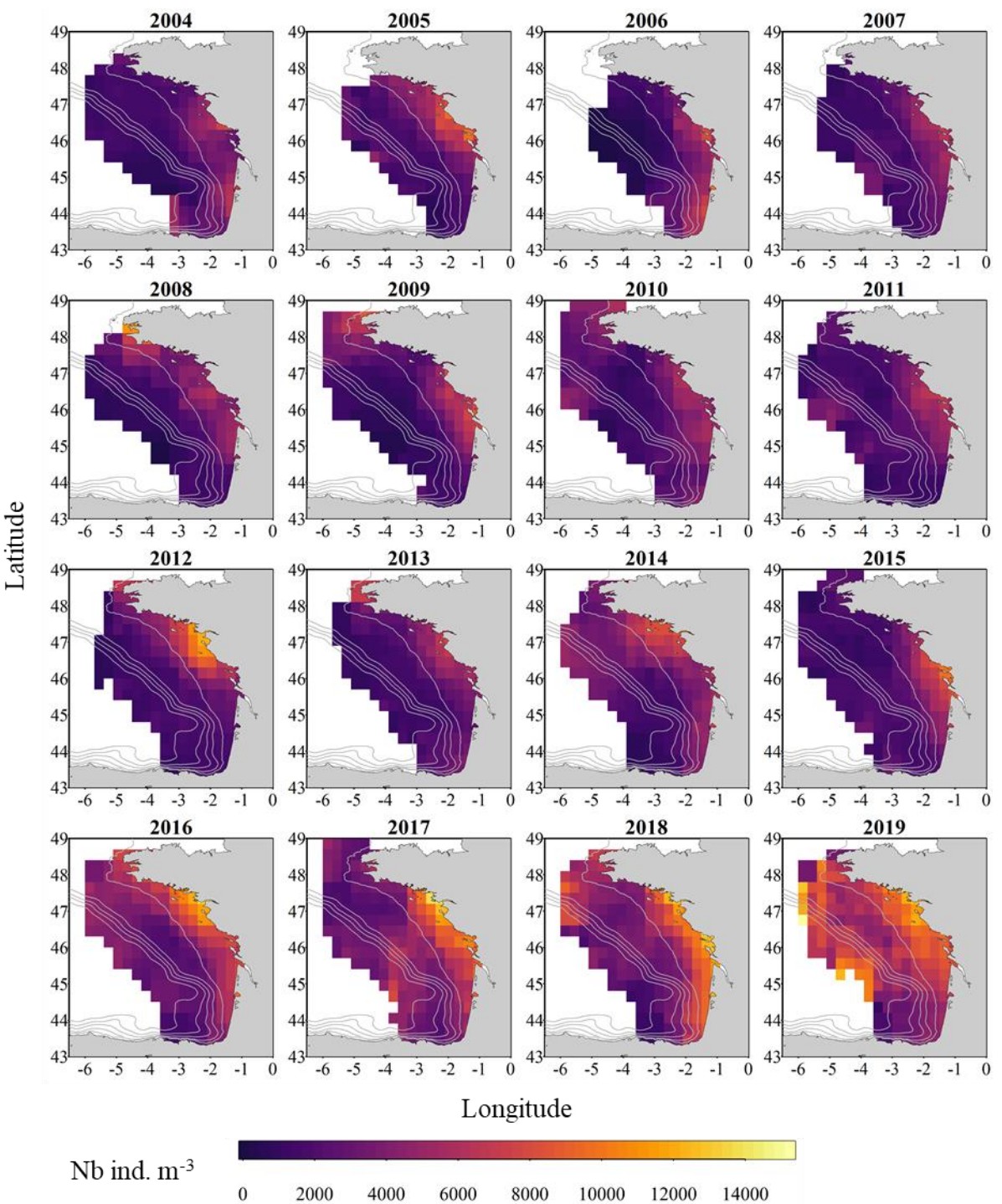

Figure 2: Gridded maps of total zooplankton abundances expressed as individuals per cubic meters of sampled seawater, during the PELGAS cruises in the Bay of Biscay from 2004 to 2019. The abundances are well within the range of zooplankton abundances seen over other temperate continental shelves. They exhibit a marked coastal to offshore gradient, abundances being higher at the coast. Abundances also show an overall increase over the years. The gridding procedure is presented in Petitgas et al. (2009) and Petitgas et al. (2014). See also Doray et al. (2018c) and Grandremy et al. (2023a) for application examples.

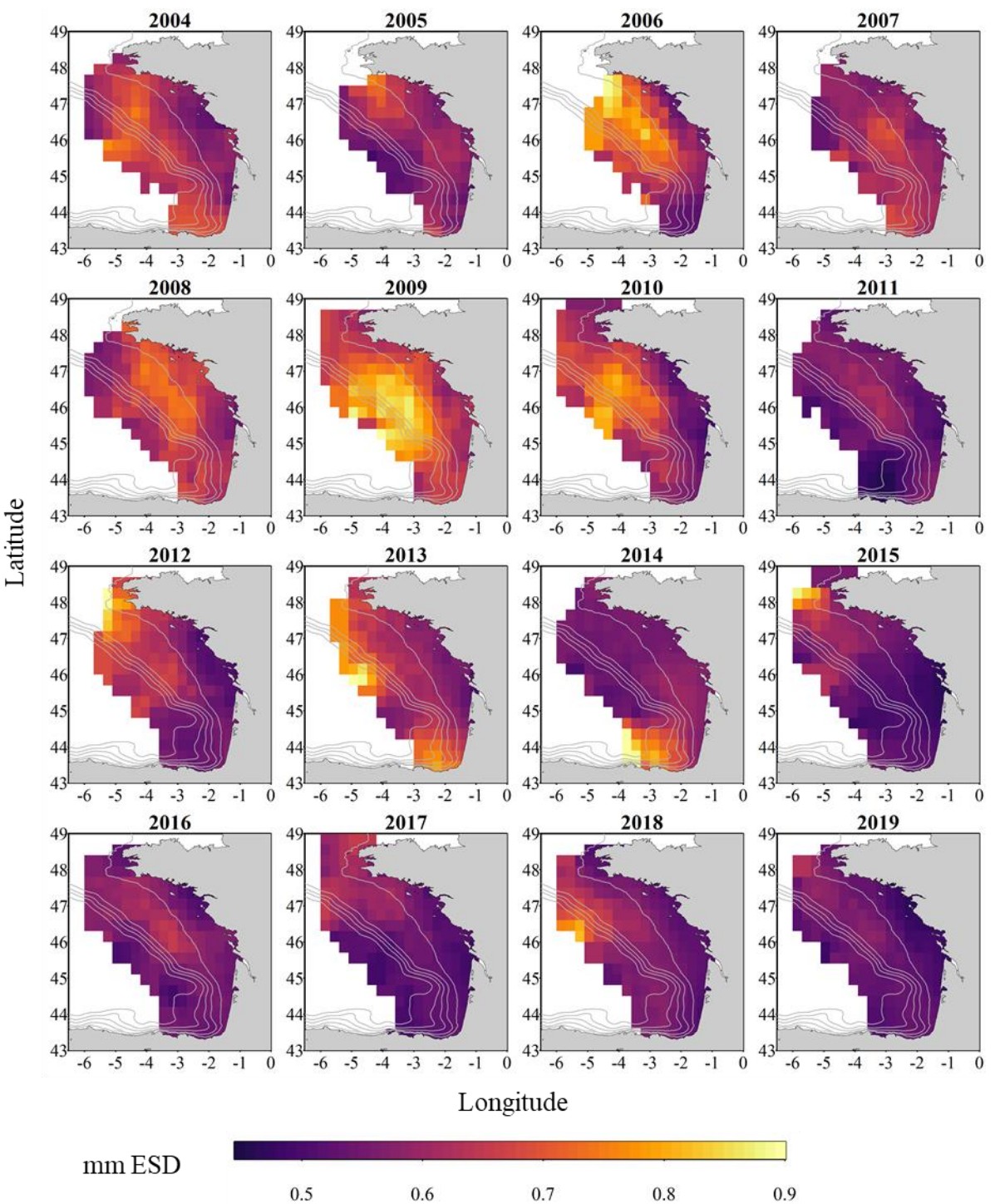

Figure 3: Gridded maps of total zooplankton mean sizes expressed as mm Equivalent Spherical Diameter during the PELGAS cruise in the Bay of Biscay from 2004 to 2019. They exhibit a coastal to offshore gradient as well as a north-south gradient. Mean body sizes are smaller at the coast and usually smaller in the south. In general, mean body sizes show an overall decrease over the years. The gridding procedure is presented in Petitgas et al. (2009) and Petitgas et al. (2014). See also Doray et al. (2018c) and Grandremy et al. (2023a) for application examples.

## 3 Datasets

### 3.1 Taxonomic groups and Operational Morphological Groups

The ZooScan dataset is composed of 1,153,507 zooplankton individuals, zooplankton parts, non-living particles and imaging artefacts individually imaged and measured with the ZooScan and ZooProcess (Gorsky et al., 2010), sorted in 127 taxonomic and morphological groups. The ZooCAM dataset is composed of 702,111 zooplankton individuals, zooplankton parts, non-living particles and imaging artefacts individually imaged and measured with the ZooCAM (Colas et al., 2018), sorted in 127 taxonomic and morphological or life stages groups. The total number of different groups identified with both instruments combined is 170, among which 84 are in common (Table 1), 43 belong to the ZooScan dataset only and 43 others belong to the ZooCAM dataset only (Table 2). The identified groups were divided into actual taxa and Operational Morphological Groups (OMGs). Typically, OMGs are either non-adult life stages of taxa, aggregated morphological groups, or non-living groups (see Tables 1 and 2). Among the groups common to both instruments, 45 are actual taxa, and 39 are OMGs (Table 1). Among the ZooScan only groups, 22 are taxa, and 21 are OMGs, and among the ZooCAM only groups, 18 are taxa, and 25 are OMGs (Table 2).

The differences in identified groups, in the ratio taxa/OMGs, and in the associated counts arose from several aspects of the data generation. Firstly, the two imaging methods differ in their technical set-up. The main difference is that, on the one hand, fixed organisms are laid down and arranged manually on the imaging sensor and digitized in a lab, steady 2-D, set-up when using the ZooScan. On the other hand, organisms are imaged live, in a moving fluid, in a 3-D environment (the flowcell), on-board when digitized with the ZooCAM. Their position in front of the camera may not enable an identification as precise as when they are laid on the scanner tray (Grandremy et al., 2023b; Colas et al., 2018). Secondly, the dataset are sequential in time, the ZooCAM dataset follows the ZooScan's. Zooplankton communities in the Bay of Biscay may have changed over time, even if their biomass as aggregated groups show a remarkable space-time stability (Grandremy et al., 2023a). Thirdly, we cannot guaranty that there is no adverse effect on taxonomic identification, as validation involved several experts (Culverhouse, 2007). Although we paid great attention to homogenize the final detailed datasets, we recommend to aggregate taxa and OMGs and reduce the biological resolution for ecological studies (Grandremy et al., 2023a, 2023b). Additionally, numerous identified and sorted taxa and OMGs do not belong to the metazoan zooplankton, or are non-adult life stages, or parts of organisms. Those were included in the presented datasets because they are always found in natural samples. They need to be separated from entire organisms to ensure as accurate as possible abundances estimations, as well as taken into account to ensure accurate biovolumes or biomasses estimations. A good example is the siphonophore issue: numerous swimming bells of degraded siphonophores individuals can be found and imaged in a sample. Determining an accurate siphonophore abundance may not be easy, but this could be overcome by considering the biovolume or biomass of siphonophores by adding up the numerous parts' biovolumes or biomass of the organisms imaged.

Table 1: ZooCAM and ZooScan common taxa and  Operational Morphological Groups (OMGs). Taxa are listed in the left column of the table, and OMGs are listed in the right column of the table . OMGs names are spelled as they appear in the dataset. Numbers next to each taxa and OMGs are the counts and the percentages (%) for each category for each instrument in the whole datasets. Non-zooplanktonic OMGs are highlighted in bold, and genera and species are formatted in italics.

| taxa | ZooCAM counts | % | ZooScan counts | % | OMG | ZooCAM counts | % | ZooScan counts | % |
|---|---|---|---|---|---|---|---|---|---|
| Calanoida | 137536 | 19.588 | 149956 | 13.00 | **detritus** | 105751 | 15.06 | 219541 | 19.03 |
| Oithonidae | 112977 | 16.09 | 110510 | 9.58 | *diatoma* | 36842 | 5.25 | 1084 | 0.09 |
| Acartiidae | 30403 | 4.33 | 66353 | 5.75 | **bubble** | 32563 | 4.64 | 1112 | 0.10 |
| Temoridae | 13520 | 1.93 | 31335 | 2.72 | **Noctiluca_Noctilucaceae** | 22165 | 3.16 | 20784 | 1.80 |
| Oncaeidae | 11843 | 1.69 | 34651 | 3.00 | other_living | 15029 | 2.14 | 5861 | 0.51 |
| Calanidae | 9578 | 1.36 | 91513 | 7.93 | dead_copepoda | 13383 | 1.91 | 17151 | 1.49 |
| Limacinidae | 8966 | 1.28 | 6423 | 0.56 | **fiber_detritus** | 13379 | 1.91 | 25124 | 2.18 |
| Appendicularia | 6724 | 0.96 | 34027 | 2.95 | nauplii_cirripedia | 6766 | 0.96 | 6008 | 0.52 |
| Cladocera | 5590 | 0.80 | 18213 | 1.58 | gonophore_diphyidae | 4395 | 0.63 | 1462 | 0.13 |
| Centropagidae | 4592 | 0.65 | 14651 | 1.27 | multiple_copepoda | 3740 | 0.53 | 961 | 0.08 |
| *Neoceratium* | 2984 | 0.43 | 4830 | 0.42 | nauplii_crustacea | 3422 | 0.49 | 10747 | 0.93 |
| Euchaetidae | 2643 | 0.38 | 12957 | 1.12 | **artefact** | 2643 | 0.38 | 60718 | 5.26 |
| Metridinidae | 2333 | 0.33 | 15081 | 1.31 | multiple_other | 1928 | 0.27 | 10303 | 0.89 |
| Corycaeidae | 2021 | 0.29 | 4720 | 0.41 | pluteus_echinodermata | 1623 | 0.23 | 1441 | 0.12 |
| *Euterpina* | 1043 | 0.15 | 2870 | 0.25 | calyptopsis_euphausiacea | 1396 | 0.20 | 3246 | 0.28 |
| Euphausiacea | 889 | 0.13 | 1195 | 0.10 | bivalvia_mollusca | 1324 | 0.19 | 3766 | 0.33 |
| *Calocalanus* | 820 | 0.12 | 1196 | 0.10 | bract_diphyidae | 1315 | 0.19 | 386 | 0.03 |
| Chaetognatha | 624 | 0.09 | 7274 | 0.63 | cypris | 862 | 0.12 | 2363 | 0.20 |
| Harpacticoida | 481 | 0.07 | 1697 | 0.15 | nectophore_diphyidae | 839 | 0.12 | 14389 | 1.25 |
| *Obelia* | 459 | 0.07 | 1016 | 0.09 | **egg_actinopterygii** | 768 | 0.11 | 3596 | 0.31 |
| Annelida | 256 | 0.04 | 2434 | 0.21 | tail_appendicularia | 753 | 0.11 | 11349 | 0.98 |
| Decapoda | 173 | 0.02 | 471 | 0.04 | cyphonaute | 684 | 0.10 | 2218 | 0.19 |
| *Microsetella* | 116 | 0.02 | 1169 | 0.10 | eudoxie_diphyidae | 501 | 0.07 | 69 | 0.01 |
| Phoronida | 90 | 0.01 | 163 | 0.01 | larvae_echinodermata | 483 | 0.07 | 2200 | 0.19 |
| Actinopterygii | 85 | 0.01 | 2113 | 0.18 | part_siphonophorae | 279 | 0.04 | 12976 | 1.12 |
| Candaciidae | 70 | 0.01 | 2773 | 0.24 | larvae_annelida | 244 | 0.03 | 708 | 0.06 |
| Amphipoda | 68 | 0.01 | 853 | 0.07 | **egg sac_egg** | 152 | 0.02 | 394 | 0.03 |
| Tomopteridae | 58 | 0.01 | 618 | 0.05 | zoea_decapoda | 151 | 0.02 | 1405 | 0.12 |
| Ostracoda | 55 | 0.01 | 341 | 0.03 | cnidaria_metazoa | 148 | 0.02 | 4974 | 0.43 |
| Doliolida | 26 | < 0.01 | 128 | 0.01 | larvae_porcellanidae | 127 | 0.02 | 2838 | 0.25 |
| Echinodermata | 24 | < 0.01 | 253 | 0.02 | nectophore_physonectae | 106 | 0.02 | 696 | 0.06 |
| Aetideidae | 15 | < 0.01 | 75 | 0.01 | ctenophora_metazoa | 94 | 0.01 | 126 | 0.01 |
| *Branchiostoma* | 15 | < 0.01 | 210 | 0.02 | **egg unkn temp_Engraulidae temp** | 61 | 0.01 | 192 | 0.02 |
| Thecosomata | 15 | < 0.01 | 59 | 0.01 | part_ctenophora | 30 | < 0.01 | 319 | 0.03 |
| Heterorhabdidae | 8 | < 0.01 | 205 | 0.02 | tornaria larvae | 21 | < 0.01 | 83 | 0.01 |
| Pontellidae | 6 | < 0.01 | 299 | 0.03 | **egg_other** | 17 | < 0.01 | 2281 | 0.20 |
| Cumacea | 4 | < 0.01 | 180 | 0.02 | megalopa | 6 | < 0.01 | 460 | 0.04 |
| Mysida | 3 | < 0.01 | 885 | 0.08 | **scale** | 2 | < 0.01 | 53 | < 0.01 |
| Eucalanidae | 2 | < 0.01 | 839 | 0.07 | siphonula | 1 | < 0.01 | 20 | < 0.01 |
| Insecta | 2 | < 0.01 | 3 | < 0.01 | | | | | |
| Foraminifera | 1 | < 0.01 | 384 | 0.03 | | | | | |
| *Haloptilus* | 1 | < 0.01 | 5 | < 0.01 | | | | | |
| Isopoda | 1 | < 0.01 | 123 | 0.01 | | | | | |
| Rhincalanidae | 1 | < 0.01 | 127 | 0.01 | | | | | |
| Sapphirinidae | 1 | < 0.01 | 21 | < 0.01 | | | | | |

Table 2: ZooCAM and ZooScan not common taxa and Operational Morphological Groups (OMGs). Taxa and OMGs appearing exclusively in the ZooCAM dataset are listed in the left column, those appearing exclusively in the ZooScan dataset are listed in the right column. OMGs names are spelled as they appear in the dataset. Numbers next to each taxa and OMG are the counts and the percentages (%) for each category for each instrument in the whole datasets. Non-zooplanktonic taxa and OMGs are highlighted in bold, and genera and species are formatted in italics.

| ZooCAM | | | ZooScan | | |
|---|---|---|---|---|---|
| taxa/OMG | counts | % | taxa/OMG | counts | % |
| **light_detritus** | 38126 | 5.43 | **badfocus_artefact** | 34507 | 2.99 |
| Rhizaria | 13347 | 1.90 | badfocus_Copepoda | 11656 | 1.01 |
| Copepoda X | 6727 | 0.96 | Eumalacostraca | 9815 | 0.85 |
| **fluffy_detritus** | 3589 | 0.51 | part_Crustacea | 7530 | 0.65 |
| *Evadne* | 1889 | 0.27 | Fritillariidae | 3635 | 0.32 |
| Hydrozoa | 1674 | 0.24 | trunk_appendicularia | 1210 | 0.10 |
| Poecilostomatoida | 1094 | 0.16 | *Aglaura* | 1113 | 0.10 |
| Rhizaria X | 857 | 0.12 | *Pleuromamma* | 695 | 0.06 |
| **Rhizosolenids** | 761 | 0.11 | part_Cnidaria | 692 | 0.06 |
| dead_harpacticoida | 528 | 0.08 | zoea_galatheidae | 660 | 0.06 |
| gelatinous | 348 | 0.05 | pluteus_ophiuroidea | 640 | 0.06 |
| ***Trichodesmium*** | 265 | 0.04 | Salpida | 470 | 0.04 |
| **aggregata** | 253 | 0.04 | Harosa | 374 | 0.03 |
| **feces** | 227 | 0.03 | tail_chaetognatha | 251 | 0.02 |
| *Halosphaera* | 193 | 0.03 | *Euchirella* | 239 | 0.02 |
| *Podon* | 162 | 0.02 | protozoea_mysida | 229 | 0.02 |
| Diphyidae | 144 | 0.02 | *Solmundella bitentaculata* | 178 | 0.02 |
| larvae_gastropoda | 116 | 0.02 | Peltidiidae | 133 | 0.01 |
| **chainlarge** | 114 | 0.02 | *Liriope tetraphylla* | 121 | 0.01 |
| veliger | 113 | 0.02 | part_Annelida | 121 | 0.01 |
| egg 1 temp_Sardina temp | 100 | 0.01 | larvae_crustacea | 114 | 0.01 |
| egg 1 temp_Engraulidae temp | 65 | 0.01 | larvae_mysida | 73 | 0.01 |
| Isias | 51 | 0.01 | ephyra_scyphozoa | 64 | 0.01 |
| egg 2 3 temp_Sardina temp | 49 | 0.01 | actinula_hydrozoa | 49 | < 0.01 |
| Calycophorae | 30 | < 0.01 | part_thaliacea | 44 | < 0.01 |
| egg 9 11 temp_Sardina temp | 26 | < 0.01 | *Atlanta* | 43 | < 0.01 |
| egg unkn temp_Sardina temp | 23 | < 0.01 | like_laomediidae | 36 | < 0.01 |
| *Calocalanus tenuis* | 17 | < 0.01 | Nemertea | 31 | < 0.01 |
| egg 4 6 temp_Sardina temp | 15 | < 0.01 | protozoea_penaeidae | 28 | < 0.01 |
| egg 9 11 temp_Engraulidae temp | 14 | < 0.01 | Cavoliniidae | 21 | < 0.01 |
| egg 7 8 temp_Engraulidae temp | 13 | < 0.01 | Actiniaria | 13 | < 0.01 |
| Enteropneusta_Hemichordata | 12 | < 0.01 | pilidium_nemertea | 12 | < 0.01 |
| ***Chaetoceros sp.*** | 9 | < 0.01 | protozoea_sergestidae | 12 | < 0.01 |
| head_crustacea | 9 | < 0.01 | phyllosoma | 8 | < 0.01 |
| *Centropages hamatus* | 8 | < 0.01 | Creseidae | 7 | < 0.01 |
| Thaliacea | 7 | < 0.01 | Penaeoidea | 7 | < 0.01 |
| egg 4 6 temp_Engraulidae temp | 6 | < 0.01 | Paguridae | 4 | < 0.01 |
| Sphaeronectidae | 4 | < 0.01 | larvae_squillidae | 4 | < 0.01 |
| *Thalassionema* | 4 | < 0.01 | Cephalopoda | 3 | < 0.01 |
| egg 2 3 temp_Engraulidae temp | 3 | < 0.01 | *Cymbulia peroni* | 3 | < 0.01 |
| *Jaxea* | 2 | < 0.01 | Nannosquillidae | 2 | < 0.01 |
| *Pyrosoma* | 1 | < 0.01 | *Lubbockia* | 1 | < 0.01 |
| larvae_ascidiacea | 1 | < 0.01 | Monstrilloida | 1 | < 0.01 |

OMGs' names are mainly in the form of two words separated by a "<" character. Although we tried to name them
as most explicitly as possible, a few potentially needed clarifications can be found in Table 3.

Table 3: Non-exhaustive list of prefixes, their types (morphological, developmental stage, taxonomical, non-living
and imaging artefact), and content.

| prefix | type | content of category |
|---|---|---|
| bract | morphological | single siphonophorae bracts |
| eudoxie | morphological | single siphonophorae eudoxia zooids |
| gonophore | morphological | single siphonophorae gonozooids |
| nectophore | morphological | single siphonophorae swimming bells |
| trunk | morphological | single appendicularian trunks detached from their tails |
| tail | morphological | appendicularian's or chaetognath's tail shaped part of the body |
| head | morphological | individual organisms' heads detached from the body |
| part | morphological | unidentified body part |
| egg sac | morphological | detached copepod egg sacs |
| like | morphological | look alike, without absolute certainty |
| multiple | morphological | two or more objects touching each other in the same vignette |
| other | morphological | non-identified living object |
| actinula | developmental stage | undefined hydrozoa actinula larval stage |
| calyptopsis | developmental stage | Euphausiacea calyptopsis larval stage |
| egg | developmental stage | egg larval stage |
| ephyra | developmental stage | ephyra hydrozoa larval stage |
| larvae | developmental stage | undefined larval stage |
| nauplii | developmental stage | crustacean nauplii larval stage |
| pilidium | developmental stage | free-swimming larvae of nemertean worm |
| protozoea | developmental stage | crustacean protozoea larval stage |
| pluteus | developmental stage | Echinodermata pluteus larval stage |
| zoea | developmental stage | crustacean zoea larval stage |
| egg 1 temp | developmental stage | clupeid fish embryo developmental stage 1* |
| egg 2 3 temp | developmental stage | clupeid fish embryo developmental stages 2 and 3 aggregated* |
| egg 4 6 temp | developmental stage | clupeid fish embryo developmental stages 4 to 6 aggregated* |
| egg 7 8 temp | developmental stage | clupeid fish embryo developmental stages 7 and 8 aggregated* |
| egg 9 11 temp | developmental stage | clupeid fish embryo developmental stages 9 to 11 aggregated* |
| egg unknown | developmental stage | clupeid fish unidentified embryo developmental stage* |
| Bivalvia | taxonomical | small bivalve larvae of unidentified mollusca |
| dead | non-living | copepod's exuvia, carcass or part of dead body |
| fiber | non-living | fiber like detritus |
| fluffy | non-living | very porous detritic particles |
| light | non-living | very transparent detritic particles |
| badfocus | imaging artefact | out-of-focus objects |

* clupeids fish embryo developmental stages according to Ahlstrom (1943) and Moser & Ahlstrom (1985).

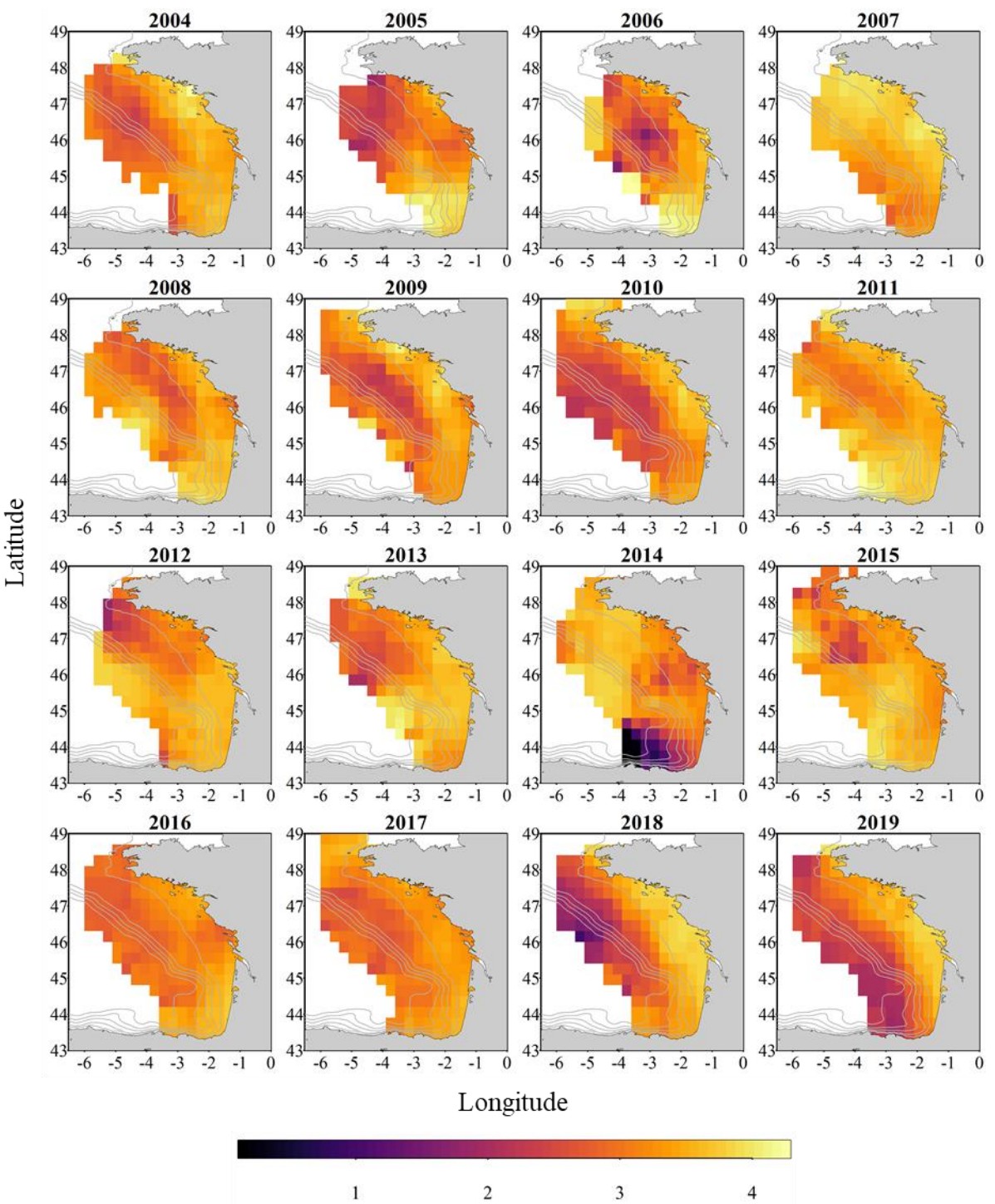

Figure 4: Gridded maps of total zooplankton Shannon index (calculated on spherical biovolumes) during the PELGAS cruise in the Bay of Biscay from 2004 to 2019. Shannon index exhibit a coastal to offshore gradient as well as a north-south gradient. Shannon index is larger at the coast and in the south, except in 2014 where it is smaller in the south, offshore. The gridding procedure is presented in Petitgas et al. (2009) and Petitgas et al. (2014). See also Doray et al. (2018c) and Grandremy et al. (2023a) for application examples.

## 3.2 Data and images

### 3.2.1 Data

The data is divided into two datasets available as tab separated files, one for each instrument. Within each dataset the data is organized as a table containing text data as well as numerical data. Each dataset combines together actual data and metadata at the individual object granularity. For each object, the user will be able to find descriptors originating from the image processing (i.e. features), and sampling metadata (i.e. latitude and longitude of sampling station, date and time of sampling, sampling device, etc.) and sample processing metadata (i.e. subsampling factor, seawater sampled volume, pixel size), in columns, and individual objects in lines. The columns headers are defined in Tables A1 and A2 for ZooCAM and ZooScan datasets respectively. The following prefixes enable the segregation of types of data and metadata: (i) "object_", which identifies variables assigned to each object individually; (ii) "sample_", which identifies variables assigned to each sample; (iii) "acq_", which identifies variables assigned to each data acquisition for the same sample (note here that this type of variable is found only in the ZooScan dataset as ZooScan samples were splitted in two size fractions corresponding to two acquisitions); (iv) "process_", which identifies variables describing key image processing features (i.e. pixel size). Those prefixes originate from the use of the Ecotaxa web application to sort and identify the images (Picheral et al., 2017) that promote this specific formatting. The ZooCAM dataset is shaped as a 72 columns (variables) x 702,111 rows (individual imaged objects) matrix and the ZooScan dataset is shaped as a 71 columns (variables) x 1,153,507 rows (individual imaged objects) matrix.

Among the 70+ variables it is worth noticing the following ones:

(i)     objid: it is a unique individual object numerical identifier that enables to link single data line to a corresponding single image in the image dataset;

(ii)    taxon: it is the taxonomic or OMG identification of the imaged objects written as they appear in the Tables 1 and 2;

(iii)   lineage: it is the full taxonomic lineage of the taxon. Lineage may be used to aggregate taxa at a higher taxonomic levels, respecting taxonomic lineages;

(iv)    classif_id: it is a unique, numerical, taxon identifier;

(v)     sample_sub_part / acq_sub_part: those are the subsampling ratios, for ZooCAM and ZooScan respectively, needed to reconstruct the quantitative estimates of the samples' abundances;

(vi)    sample_fishingvolume / sample_tot_vol: those are the total seawater sampled volumes for ZooCAM and ZooScan respectively, needed to normalize the samples' concentrations by seawater volume.

One can therefore calculate quantitative abundances estimates for a taxon in a sample as follow:

ZooCAM: $Ab_{taxon} = \frac{n_{taxon} \times sample\_sub\_part}{sample\_fishingvolume}$ (1)

ZooScan: $Ab_{taxon} = \frac{\left(n_{taxon_{acq1}} \times acq\_sub\_part_{acq1}\right) + \left(n_{taxon_{acq2}} \times acq\_sub\_part_{acq2}\right)}{sample\_tot\_vol}$ (2)

Where $Ab$ is the abundance in ind.m$^{-3}$ and $n$ is the number of individuals for "taxon".

### 3.2.2 Images

Two sets of individual images sorted into folders by categories (Tables 1 and 2) come along with each dataset. For the ZooCAM only, the associated images from years 2016 and 2017 contain printed Region Of Interest (ROI) bounding box limits and text at the bottom of each image, and non-homogenised background within and around the ROI bounding box; images from year 2018 contain non-homogenised background within the ROI bounding box only; images from 2019 have a completely homogeneous and thresholded background around the object. The differences arose from successive ZooCAM software updates that do not modify the calculation of object's features. The ZooScan images have all a completely homogeneous and thresholded background around the object, no bounding box limits nor text printed in the images. All images for the two instruments datasets have a 1 mm scale bar printed at the bottom left corner.

## 4 Data availability

The ZooScan dataset can be found as the ***PELGAS Bay of Biscay ZooScan zooplankton Dataset (2004-2016)*** in the SEANOE dataportal following the link: https://www.seanoe.org/data/00829/94052/ (Grandremy et al., 2023c). Individual objects images can be freely viewed and explored by anyone using the Ecotaxa (https://ecotaxa.obs-vlfr.fr/) web application, without registration, under the tab "explore images", by searching the project name: "***PELGAS Bay of Biscay ZooScan zooplankton Dataset (2004-2016)".***

The ZooCAM dataset can be found as the ***PELGAS Bay of Biscay ZooCAM zooplankton Dataset (2016-2019)*** in the SEANOE dataportal https://www.seanoe.org/data/00828/94040/ (Grandremy et al., 2023d). Individual objects images can be freely viewed and explored by anyone using the Ecotaxa (https://ecotaxa.obs-vlfr.fr/) web application, without registration, under the tab "explore images", by searching the project name: "***PELGAS Bay of Biscay ZooCAM zooplankton Dataset (2016-2019)".***

Each dataset comes as a .zip archive that contains:

- One tab separated file containing all data and metadata associated to each imaged and identified object.
- One comma separated file containing the name, type, definition and unit of each field (column)
- One comma separated file containing the taxonomic list of the dataset, with counts and nature of the content of the category
- A directory "*individual_images*" containing images of each object, named according to the object id *objid* and sorted in subdirectories according to their taxonomic identification, across years and sampling stations.

## 5 Concluding remarks

Recent studies showed that the small pelagic fish (SPF) communities have suffered from a drastic decrease of condition in the Mediterranean Sea and in the Bay of Biscay (Van Beveren et al., 2014; Doray et al., 2018d; Saraux et al., 2019) over the last 20 years. This loss of condition was especially expressed by the constant decrease of SPF size- and weight-at-age (Doray et al., 2018d; Veron et al. 2020), and possibly explained by a change in SPF trophic resource composition, size and quality (Brosset et al., 2016; Queiros et al., 2019; Menu et al., 2023). Identifying and measuring zooplankton at appropriate temporal and spatial scales is not an easy task, but can be addressed with imaging. These datasets were assembled as an effort to make possible the exploration

of the relationship between SPF observed dynamics in the Bay of Biscay and their main food resource's dynamics,
the metazoan zooplankton. This zooplankton imaging data series is a significant output of Nina Grandremy PhD
(2019-2023), that is currently being exploited (Grandremy et al., 2023a), and is intended to be continued and
updated on a yearly basis in the framework of the PELGAS program, to better understand the underlying processes
presiding to long-term SPF dynamics. Moreover, those two zooplankton datasets can be associated with the
PELGAS survey datasets previously published in 2018, also in the SEANOE dataportal, featuring hydrological,
primary producers, fish and megafauna data arranged as gridded data (Doray et al., 2018b). Together, all these
datasets allow to study simultaneously all the pelagic ecosystem compartments, with coherent spatial domain (the
Bay of Biscay continental shelf), resolution and time series. Nevertheless, a spatial gridding of the data is highly
recommended (as represented in the Fig. 2, 3 and 4), since the spatial coverage of the sampling protocols can vary
between years (Fig. 1), within and between each pelagic ecosystem compartment. A procedure for such batch data
spatial smoothing is presented e.g. in Petitgas et al. (2009) and Petitgas et al. (2014). See also Doray et al. (2018c)
and Grandremy et al. (2023a) for application examples. As several descriptors of the spring zooplankton
community (abundances, sizes, biovolumes, biomass) can be derived from this 16 years long spatially resolved
time series at several taxonomic levels, these datasets are intended to be used in various ecological studies
including the zooplankton compartment, especially modelling studies, where zooplankton is usually
underrepresented (Mitra, 2010; Mitra et al., 2014). Finally, these datasets can also be used for machine learning
applied to plankton studies serving, for example, as consequent learning sets.

## Disclaimer

Data are published without any warranty, express or implied. The user assumes all risk arising from his/her use of
data. Data are intended to be research-quality, but it is possible that the data themselves contain errors. It is the
sole responsibility of the user to assess if the data are appropriate for his/her use, and to interpret the data
accordingly. Authors welcome users to ask questions and report problems.

## Authors' contributions

GN scanned and validated most of the ZooScan dataset, assembled the datasets, and led the drafting. BP collected
and managed the samples since 2004, and participated in the manual validation of identifications. DE scanned a
substantial fraction of the ZooScan samples and participated in the initial sorting of vignettes. DMM participated
in the collection of samples, and was involved in the ZooCAM development. DM was chief scientist on the
PELGAS surveys and participated in the drafting. DC supervised GN work and participated in the drafting. FB
developed, improved and maintained the ZooCAM software. JL curated a substantial fraction of the ZooScan
dataset manual validation of identifications. HM participated in the collection of samples, lead the DEFIPEL
project, and participated in the drafting. LMS participated in the collection of samples, and managed the ZooCAM.
NA curated a substantial fraction of the ZooScan and ZooCAM dataset manual validation of identifications. PP
supervised GN work and participated in the drafting. PPh participated in the collection of samples and participated
in the drafting. RJ supervised the development and improvement of the ZooCAM. TM developed and improved
the ZooCAM, and participated in the collection of samples. RJB supervised GN work, participated in the collection
of samples, curated a substantial fraction of the ZooCAM dataset manual validation of identifications, and lead
the drafting.

## Competing interests

The authors declare that they have no conflict of interest.

## Acknowledgements

The authors acknowledge receiving funding from the 'France Filière Pêche' DEFIPEL project. NG acknowledges the funding of her PhD by Region Pays de la Loire, FR and Ifremer. The authors wish to thank Jean-Yves Coail, Gérard Guyader and Patrick Berriet (Ifremer – REM-RDT-SIIM) for their contribution to the hardware assembly of the ZooCAM. The authors acknowledge the work of Elio Raphalen for scanning year 2005 samples. The authors thank the EMBRC platform PIQs for image analysis. This work was supported by EMBRC-France, whose French state funds are managed by the ANR within the Investments of the Future program under reference ANR-10-INBS-02. Finally, the authors wish also to thank the many other students, technicians and scientists who participated in the sampling and samples imaging on board, and the successive crews of the R/V *Thalassa* involved in the PELGAS surveys from 2004 to 2019.

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

# Appendix A

Table A1: ZooCAM dataset columns header – definition of data and metadata fields.

| column name | definition |
| --- | --- |
| object_id | name of object and associated image |
| objid | unique ecotaxa internal object identifier |
| object_lat | latitude of sampling |
| object_lon | longitude of sampling |
| object_date | date of sampling |
| object_time | time of sampling |
| object_depth_min | minimum sampling depth |
| object_depth_max | maximum sampling depth |
| object_taxon | taxonomic name |
| object_lineage | full taxonomic lineage corresponding to the taxon |
| classif_id | unique ecotaxa internal taxon identifier |
| object_area | object's surface |
| object_area_exc | object surface excluding white pixels |
| object_%area | proportion of the image corresponding to the object |
| object_area_based_diameter | object's Area Based Diameter: 2 * (object_area/pi)^(1/2) |
| object_meangreyimage | mean image grey level |
| object_meangreyobjet | mean object grey level |
| object_modegreyobjet | modal object grey level |
| object_sigmagrey | object grey level standard deviation |
| object_mingrey | minimum object grey level |
| object_maxgrey | maximum object grey level |
| object_sumgrey | object grey level integrated density: object_mean*object_area |
| object_breadth | breadth of the object along the best fitting ellipsoid minor axis |
| object_length | breadth of the object along the best fitting ellipsoid majorr axis |
| object_elongation | elongation index: object_length/object_breadth |
| object_perim | object's perimeter |
| object_minferetdiam | minimum object's feret diameter |
| object_maxferetdiam | maximum object's feret diameter |
| object_meanferetdiam | average object's feret diameter |
| object_feretelongation | elongation index: object_maxferetdiam/object_minferetdiam |
| object_compactness | Isoperimetric quotient: the ratio of the object's area to the area of a circle having the same perimeter |
| object_intercept0 | number of times that a transition from background to foreground occurs a the angle 0° for the entire object |
| object_intercept45 | the number of times that a transition from background to foreground occurs a the angle 45° for the entire object |
| object_intercept90 | the number of times that a transition from background to foreground occurs a the angle 90° for the entire object |
| object_intercept135 | the number of times that a transition from background to foreground occurs a the angle 135° for the entire object |
| object_convexhullarea | area of the convex hull of the object |
| object_convexhullfillratio | ratio object_area/convexhullarea |
| object_convexperimeter | perimeter of the convex hull of the object |
| object_n_number_of_runs | number of horizontal strings of consecutive foreground pixels in the object |
| object_n_chained_pixels | number of chained pixels in the object |
| object_n_convex_hull_points | number of summits of the object's convex hull polygon |
| object_n_number_of_holes | number of holes (as closed white pixel area) in the object |
| object_transparence | ratio object_sumgrey/obejct_area |
| object_roughness | measure of small scale variations of amplitude in the object's grey levels |
| object_rectangularity | ratio of the object's area over its best bounding rectangle's area |
| object_skewness | skewness of the object's grey level distribution |
| object_kurtosis | kurtosis of the object's grey level distribution |
| object_fractal_box | fractal dimension of the object's perimeter |
| object_hist25 | grey level value at quantile 0.25 of the object's grey levels normalized cumulative histogram |
| object_hist50 | grey level value at quantile 0.5 of the object's grey levels normalized cumulative histogram |
| object_hist75 | grey level value at quantils 0.75 of the object's grey levels normalized cumulative histogram |
| object_valhist25 | sum of grey levels at quantile 0.25 of the object's grey levels normalized cumulative histogram |
| object_valhist50 | sum of grey levels at quantile 0.5 of the object's grey levels normalized cumulative histogram |
| object_valhist75 | sum of grey levels at quantile 0.75 of the object's grey levels normalized cumulative histogram |
| object_nobj25 | number of objects after thresholding at the object_valhist25 grey level |
| object_nobj50 | number of objects after thresholding at the object_valhist50 grey level |
| object_nobj75 | number of objects after thresholding at the object_valhist75 grey level |
| object_symetrieh | index of horizontal symmetry |
| object_symetriev | index of vertical symmetry |
| object_thick_r | maximum object's thickness/mean object's thickness |
| object_cdist | distance between the mass and the grey level object's centroids |
| object_bord | tag for object touching the frame edge |
| sample_id | name of the sample from where the object originates |
| sample_ship | name of the ship used to collect the samples |
| sample_campaign | name of the cruise where samples were collected |
| sample_station | name of the station where samples were collected |
| sample_depth | bottom depth at station |
| sample_device | net used to collect the sample |
| sample_fishingvolume | seawater volume sampled |
| sample_sub_part | subsampling elevation factor |
| process_id | name of software/software version used to analysed digitized sample images |
| process_resolution_camera_micron_ | pixel size, µm |

Table A2: ZooScan dataset columns header – definition of data and metadata fields

| column name | definition |
| --- | --- |
| object_id | name of object and associated image |
| objid | unique ecotaxa internal object identifier |
| object_lat | latitude of sampling |
| object_lon | longitude of sampling |
| object_date | date of sampling |
| object_time | time of sampling |
| object_depth_min | minimum sampling depth |
| object_depth_max | maximum sampling depth |
| object_taxon | taxonomic name |
| object_lineage | full taxonomic lineage corresponding to the taxon |
| classif_id | unique ecotaxa internal taxon identifier |
| object_area | object's surface |
| object_mean | mean object grey level |
| object_stddev | object grey level standard deviation |
| object_mode | modal object grey level |
| object_min | minimum object grey level |
| object_max | maximum object grey level |
| object_perim. | object's perimeter |
| object_major | lenght of major axis of best fitting elipse |
| object_minor | lenght of minor axis of best fitting elipse |
| object_circ. | circularity: 4*pi(object_area/object_perim.^2) |
| object_feret | maximum feret diameter |
| object_intden | object grey level integrated density : /object_mean*/object_area |
| object_median | median object grey level |
| object_skew | skewness of the object's grey level distribution |
| object_kurt | kurtosis of the object's grey level distribution |
| object_%area | proportion of the image corresponding to the object |
| object_area_exc | object surface excluding white pixels |
| object_fractal | fractal dimension of the object's perimeter |
| object_skelarea | surface of the one-pixel wide skeleton of the object |
| object_slope | slope of the cumulated histogram of the object grey levels |
| object_histcum1 | the number of times that a transition from background to foreground occurs at the angle0° |
| object_histcum2 | grey level at quantiles 0.5 of the histogram of the object grey levels |
| object_histcum3 | grey level at quantiles 0.75 of the histogram of the object grey levels |
| object_nb1 | number of objects after thresholding at the object_histcum1 grey level |
| object_nb2 | number of objects after thresholding at the object_histcum2 grey level |
| object_symetrieh | index of horizontal symmetry |
| object_symetriev | index of vertical symmetry |
| object_symetriehc | index of horizontal symmetry after thresholding at the object_histcum1 grey level |
| object_symetrievc | index of vertical symmetry after thresholding at the object_histcum1 grey level |
| object_convperim | perimeter of the convex hull of the object |
| object_convarea | area of the convex hull of the object |
| object_fcons | object's contrast |
| object_thickr | maximum object's thickness/mean object's thickness |
| object_esd | object's Equivalent Spherical Diameter: 2 * (object_area/pi)^(1/2) |
| object_elongation | elongation index: major/minor |
| object_range | range of greys: max-min |
| object_meanpos | relative position of the mean grey: (max-mean)/range |
| object_centroids | distance between the mass and the grey level object's centroids |
| object_cv | coefficient of variation of greys: 100*(stddev/mean) |
| object_sr | index of variation of greys: 100*(stddev/range) |
| object_perimareaexc | index of the relative complexity of the perimeter: object_perim/object_area_exc |
| object_feretareaexc | another elongation index : object_feret/object_area_exc |
| object_perimferet | index of the relative complexity of the perimeter: object_perim/object_feret |
| object_perimmajor | index of the relative complexity of the perimeter: object_perim/object_major |
| object_circex | circularity of object excluding white pixels: 4*pi(object_area_exc/object_perim.^2) |
| object_cdexc | distance between the mass and the grey level object's centroids calculated with object_area_exc |
| sample_id | name of the sample from the object originate |
| sample_ship | name of the ship used to collect the samples |
| sample_program | name of the cruise where samples were collected |
| sample_stationid | name of the station where samples were collected |
| sample_bottomdepth | bottom depth at station |
| sample_net_type | net used to collect the sample |
| sample_tot_vol | seawater volume sampled |
| sample_comment | comments associated with sampling/sample treatment |
| process_id | name of software/software version used to analysed digitized sample images |
| process_particle_pixel_size_mm | pixel size |
| acq_id | name of subsample if any |
| acq_min_mesh | minimum sieve size of subsample |
| acq_max_mesh | maximum sieve size of subsample |
| acq_sub_part | subsampling elevation factor |

595

596