# Peer review of "Metazoan zooplankton in the Bay of Biscay: 16 years of individual sizes and abundances combining ZooScan and ZooCAM imaging systems."

_Earth System Science Data, 2023_

## Author Response (AR1)

Nina GRANDREMY
grandremy.n@gmail.com

To François G. Schmitt, handling topic editor in ESSD.

Dear François G. Schmitt,

We are pleased to submit our response to the reviewers and the revised version of our manuscript entitled "Metazoan zooplankton in the Bay of Biscay: 16 years of individual sizes and abundances combining ZooScan and ZooCAM imaging systems" (essd-2023-187). We would like to sincerely thank the two anonymous reviewers and Dr Leo Berline for their useful comments, and we hope that the modifications made to the manuscript following their reviews have sufficiently improve the manuscript to entail its publication in ESSD.

In brief, we acknowledged that the Tables 1 and 2 were difficult to understand because of the font italic and the term 'OTU' that were not used in the appropriate way. Both tables have been modified to take into account all the comments. Also, we added in the Methods section a more detailed description of the PELGAS survey from which the zooplankton samples come from. We improved the section about the study regarding the comparison of the ZooScan and the ZooCAM, adding some explanations about the discrepancies observed between instruments (Grandremy et al., 2023b). Finally, we corrected all the spelling mistakes and updated some references as requested.

All the comments have been answered in the ESSD Discussion and below, our responses being highlighted here in blue. We hope that the revised version of this manuscript is now suitable for publication.

On the behalf of all the co-authors, respectfully,

N. Grandremy.

**RC1: 'Comment on essd-2023-187', Anonymous Referee #1, 06 Jul 2023.**

The compilation of image datasets in the Bay of Biscay of the course of 16 year is a very useful addition to the freely available zooplankton datasets. These datasets can be widely used for ecological or modelling purposes.

General remark: I have wondered about the abbreviation of PELGAS. Maybe it would be a good idea to explain the program either in Methods or Introduction as it is a central part of the data sets.

We thank Referee #1 for the advice.

Zooplankton samples were collected during the successive PELGAS (PELagique GAScogne) integrated surveys carried out over the Bay of Biscay (BoB) French continental shelf, every year in spring from 2004 to 2019 on board the R/V Thalassa. The aim of this survey is to assess small pelagic fish biomass and monitor the pelagic ecosystem to inform ecosystem based fisheries management. Fish data, hydrology, phyto- and zoo-plankton samples and megafauna sightings (marine mammals and seabirds) are concomitantly collected to build long-term spatially resolved time series of the BoB pelagic ecosystem. The PELGAS sampling protocols combine day-time en-route data collection (small pelagic fish and megafauna), with night-time, depth integrated hydrology and plankton sampling at fixed points. Detailed PELGAS survey protocols can be found in Doray et al., 2018a and Doray et al., 2021.

This paragraph has been added to the text at the beginning of the section 2.1 "Sampling", lines 102-110.

I find the presentation of the taxa and OTUs in table 1 and 2 a bit difficult. Usually italics are reserved for genera and species names. I suggest to change that in both tables. Additionally it would be good to add the intruments in the header of table 2.

We understand that italics were not used in the appropriate way here. In the revised manuscript, this font has been kept for actual genera and species only in Tables 1 and 2. Higher order taxa (e.g.: family, orders, etc.) and Operational Morphological Groups have all been formatted in non-italics font. The instrument names have been added in the columns' header in Table 2. Please see line 285 (Table 1) and line 294 (Table 2).

I think it would also be good to add at a more prominent place that environmental and other accompanying data are availabel for the same cruise. So far as I noticed this is only mentioned in the concluding remarks. However, for the usability of these datasets environmetnal data are essential and should be highlighted somewhere earlier in the manuscript.

We thank Referee #1 for this remark. A sentence about the environmental datasets availability has been added at the beginning of the section 2.1 "Sampling", following the PELGAS survey description in the revised manuscript, please see lines 110-112.

I have a few orthographic remarks:

We thank Referee #1 for pointing out these orthographic remarks.

Line 40: … objects are composed of… Corrected at line 40.

Line 56: Metazoan planktonic organisms…. Corrected at line 56.

Line 77/78: …taxonomic analysts… Corrected at line 78.

Line 86: …these require… Corrected at line 87.

Line 189: quality check of identifications?

A subsequent quality check of automatic taxonomic identifications has been realized in a two-step process: a first complete review (validation and/or correction) of all individual automatic identifications was done by GN and RJB; then, trained experts (JL and NA) reviewed and curated the ZooScan and ZooCAM datasets, respectively, at the individual level. Although some identification errors may still remain in the datasets, we consider this double check process as sufficient to provide taxonomically qualified data.

This paragraph has been inserted in the manuscript main text, see lines 198-203.

**CC1: 'Comment on essd-2023-187', Leo Berline, 29 Sep 2023.**

This study is very useful as freely available zooplankton datasets are not very common, especially at this large scale.

I would recommend to clarify/insist on the homogeneity of the two datasets (Zooscan and ZooCAM) in the whole ms.

We appreciate this recommendation. However, the two datasets are not completely homogenous as provided. They are both as taxonomically detailed as our identification skills enabled, and although being intercomparable and interoperable for most of the taxa and most of the oceanographic conditions, there are still some discrepancies between the two instruments, particularly on the abundances estimations (Grandremy et al., 2023b). Yet, those can be easily overcome by aggregating some taxa in one dataset or the other, or both, but the resulting taxonomic resolution would be inferior to that of the provided dataset. We therefore not insist on the homogeneity, rather on the intercomparability and interoperability of the two instruments. We will however take into account your titling suggestion as it actually reflects the combination of instruments already undertaken in a previous paper (Grandremy et al., 2023a).

The title may state this: replace "from the" with "combining" ZooScan and ZooCAM imaging systems. Done, please see line 2.

- Some typos and lack of accuracy.

Abstract: L40 "are composed". Done, please see line 40.

I am not familiar to the meaning of 'interoperable'. Might be worth clarifying.

Interoperable refers to the FAIR principles (https://www.go-fair.org/fair-principles/). When generating scientific data, it is now recommended to generate them following the FAIR principles, that are: the data should be Findable, Accessible, Interoperable (easy to combine to other data), and Reusable.

L76: decent spatio-temporal scale: 'decent' is not a scientific word. Corrected, see lines 76-77.

L91: 'open' = freely available. Corrected, please see line 92.

L113: "...laboratory with the zooscan in 2019." I suggest to remove 'in 2019' which is useless but confusing with the sampling dates. Done, please see line 123.

L138: I would add "net collection" to insist on the fact that it is the same sample that can be analysed in both ways. Done, please see line 148.

L151-153: " break free from .." unclear sentence. I guess authors mean to control the maximum number of object. Corrected, see line 161.

L176: ZooCAM sofware? Yes, the ZooCAM software does not provide a digital tool enabling the separation of possible touching objects. The correction has been done line 186.

L184: "that were used" Corrected at line 194.

L191-192: "To be used together" I guess "combined" is the proper word. Corrected at line 208.

L190-200: for 2016, how were the samples splitted between Zooscan and ZooCAM?

A comparison study was done to ensure the datasets originating from each instrument are homogenous and can be combined for ecological studies (Grandremy et al., 2023b). All the zooplankton samples from year 2016 (61 sampling stations over the whole BoB continental shelf) were imaged with both instruments. This comment has been added to the text to clarify this point, please see lines 207-211.

L201 comparable: can you better quantify (correlation?) Corrected at line 217.

Fig 2, 3, 4: add reference for the gridding procedure. The gridding procedure is presented in Petitgas et al. (2009) and Petitgas et al. (2014). See also Doray et al. (2018c) and Grandremy et al. (2023a) for application examples. These references have been added in the figures' legends, please see lines 238-239 (Figure 2), lines 244-245 (Figure 3) and lines 308-309 (Figure 4).

L252 Table 1, 2. Why not sorting according to abundance? It is more meaningful than taxonomy. We agree with Dr Berline. Both tables have been sorted according to the counts. In Table 1, the sorting was done according the counts calculated from the ZooCAM data.

**RC2: 'Comment on essd-2023-187', Anonymous Referee #2, 05 Nov 2023.**

General comments:

This manuscript presents a very impressive dataset, covering all together 16 years of sampling (once a year) over a very large area of the Gulf of Biscay and operated using two separate imaging techniques.

The fact that presenting and providing the full dataset (both on the form of raw data and in the form of images) is a really large contribution to make important and precious dataset as FAIR as possible and the way the dataset is presented contributes to a clear understanding of the protocols operated, but also on the analysis conducted to obtain the final raw dataset. The fact that both datasets have been cross calibrated to provide basis on their Interoperability (the 'I' in FAIR) is also one really nice and important contribution (but is published and extensively inspected in https://doi.org/10.1002/lom3.10577 , maybe the citation and link to this latter could be made more visible), and increase its robustness.

We thank Referee #2 for the positive comment. At the time we wrote this data paper, the publication presenting the interoperability of both instruments was under review. It is now published and the citation of this publication has been updated with the DOI in the present manuscript, please see line 500-503.

In addition to the provided "raw" data (hosted in the platform SEANOE, and freely available to anyone; therefore, making it FA(i)R), I would strongly recommend to provide the "pre-analyzed" data (ie. the ones used to currently produce the figures 2,3, and 4 (abundance, size and shanon index, maybe separated by instrument types, since here it appears unclear which instrument is used for the year where both were used for intercalibration). Such additional data availability will make the dataset even more "Reusable" for users without having to recompute everything from raw data.

The figures 2, 3 and 4 in the manuscript illustrate a few examples of the zooplankton community's descriptors that can be derived from these datasets, but this list is not exhaustive. Biomass, biovolumes, size spectra can also be calculated from the data. The use of these datasets will depend on the scientific question of each study in which they will be included. The main interest of providing raw individual data is to allow users to aggregate the data at the taxonomic resolution they require for their specific scientific needs but also to use the data at the individual level as provided and / or to define functional traits based on imagery or taxonomy. Also, the figures 2, 3 and 4

show the data spatially smoothed over a grid with a cell size set at 0.3° in latitude and longitude. Providing raw individual data at each sampling stations will allow users to work at the spatial resolution needed (sampling station or spatial smoothing over grids with different cell sizes or averaged over the whole Bay of Biscay). Therefore, we think that providing the raw individual data at each sampling station, as presented here, is the easiest way to make these datasets reusable.

The fact that the raw images are also provided is also a very important feature, although providing 6Go of dataset at once will certainly limit the number of people which will actually access to this massive dataset (except for needs of machine learning techniques). In addition to this, I would provide a way to interactively access and inspect the imaging dataset without actually downloading it. Maybe providing the full list of ecotaxa projects (currently accessible from what I see, e.g. https://ecotaxa.obs-vlfr.fr/prj/2952 they seems to be separated by instrument and years of sampling) and allowing readers to get a preview and inspect the dataset could be a really nice addition, maybe as an additional table in the manuscript.

We thank Referee #2 for the idea of providing a way for the users to check the imaging dataset without necessarily downloading it. Each dataset corresponds to a project on the platform Ecotaxa that gather the data from all the years for each instrument (2004-2016 for the ZooScan and 2016-2019 for the ZooCAM). We now provide the names of both of these projects to allow the readers to browse and inspect the individual vignettes on Ecotaxa, see lines 357-359 (ZooScan dataset) and lines 361-364 (ZooCAM dataset). On the other hand, we would like to clarify that the 6 Go dataset available through SEANOE constitute the individual images only that can be downloaded separately from the numerical data. We assume that these sets of images will be useful in the machine-learning field where computers are powerful enough to handle these very large datasets.

Although convincing in GrandRemy et al 2023, the intercomparison of instruments is certainly not fully perfect. And from table 3 only, it seems that some taxonomic groups do exhibit some discrepancies in between the two datasets (clearly overseen by one instrument). This is the case for Eucalanidae, Foraminifera, Isopoda, Mysida, Pontelidae, Rhincalanidae, Sapphiridae at least. However, this intercomparison is made complicated since data provided in table 1 are "raw counts" and total counts per instruments are not equal. Eventually providing percentages would also allow a better feel on this, and maybe a more complete disclaimer on potential discrepancies at fine taxonomical resolution between the two datasets could be needed in the section 2.6… and could also be served by table 2 (e.g. rhizaria in zooscan been identified under the label harosa in zoocam).

Reviewer 2 is right, the intercomparison is not perfect, as already discussed in a comment above (Reply to Leo Berline).

The Tables 1 and 2 have been modified to show the percentage calculated for each taxa and OMGs. The discrepancies showed by some taxonomic groups between instruments originates from the differences in sample preparation protocols before the image acquisition, the imaging techniques and quality, and whether the samples were imaged live or fixed. For example, the mineralized protists (here, Rhizaria) dissolve in formalin and are considered underestimated in preserved seawater samples (Biard et al., 2016). Also, the random orientation of objects in the ZooCAM flow cell leads to a loss of taxonomic identification accuracy due to the difficulty to spot the specific features needed for the identification (Colas et al., 2018; Grandremy et al., 2023b). This is particularly acute for copepods, where the ZooScan seems to provide better identification capabilities to experts, as the organisms are imaged in a lateral view most of the time whereas the ZooCAM often images them in a non-lateral, randomly-oriented view, preventing the visualisation of specific features. A detailed discussion about how to explain the discrepancies between the ZooScan and the ZooCAM can be found in Grandremy et al. (2023b).

This paragraph has been added at the end of the section 2.6, lines 220 to 230.

One modification that should also been conducted is the probably very incorrect usage of the term OTU to defines the different taxonomic groups used here. While it may correspond to the initial definition of OTU (operational grouping of closely related individuals…. Here by morphological properties), this latter is currently too much used by genomic approaches to not bring to confusions. Additionally, here some "OTU" are used to define non biological entities (therefore not having any "taxonomy", such as bubbles or detritus), making even the initial definition of OTU non pertinent. More importantly with the initial definition of what is called "taxa", those are indeed the real OTU (following the initial definition). All this to show that the usage of the term OTU is definitively confusing and I would strongly advice to not use this term and use a more adequate one (morphological groups, "Operational Morphological groups" or any more pertinent terms).

We understand that the term OTU is confusing. It has been replaced by the term "Operational Morphological Groups" (OMG) as this category combined aggregated morphological groups, non-adult life stages of taxa and non-living objects groups (e.g.: detritus, artefacts). The change has been made throughout the manuscript, mostly in the section 3.1 and in Tables 1 and 2.

Detailed comment:

Table 2 seems inverted in the columns: the title state that we are inspecting Zooscan and Zoocam data while the rest of the caption seems to indicate that the table is oriented on showing Zoocam (left) and Zooscan (right) (therefore inverted into order). Since the two sides of the table have no labels, this makes the interpretation somewhat confusing. I would advise to keep the same order (and add labels to the table).

We thank Referee #2 for pointing out the confusion on Table 2. As stated in the table's caption, taxa and OMGs appearing exclusively in the ZooCAM dataset are listed in the left column while those appearing exclusively in the ZooScan dataset are listed in the right column, following the same order as in Table 1. The title of Table 2 caption have been modified to "ZooCAM and ZooScan not common taxa and OMG." and the instrument names have been added to the table's header for more clarity. Please see lines 288-294.

Table 3: ephyra not ephira Corrected, please see Table 3, line 300.